# Public opinion on sharing data from health services for clinical and research purposes without explicit consent: an anonymous online survey in the UK

Linda A Jones [ORCID],[1] Jenny R Nelder,[1] Joseph M Fryer,[1] Philip H Alsop,[2] Michael R Geary,[2] Mark Prince,[2] Rudolf N Cardinal [ORCID] [1,3]

¹Department of Psychiatry, University of Cambridge, Cambridge, UK
²Cambridge, UK
³Liaison Psychiatry Service, Cambridgeshire and Peterborough NHS Foundation Trust, Cambridge, UK

**Correspondence to**
Dr Rudolf N Cardinal;
rnc1001@cam.ac.uk

## ABSTRACT

**Objectives** UK National Health Service/Health and Social Care (NHS/HSC) data are variably shared between healthcare organisations for direct care, and increasingly de-identified for research. Few large-scale studies have examined public opinion on sharing, including of mental health (MH) versus physical health (PH) data. We measured data sharing preferences.

**Design/setting/interventions/outcomes** Pre-registered anonymous online survey, measuring expressed preferences, recruiting February to September 2020. Participants were randomised to one of three framing statements regarding MH versus PH data.

**Participants** Open to all UK residents. Participants numbered 29 275; 40% had experienced an MH condition.

**Results** Most (76%) supported identifiable data sharing for direct clinical care without explicit consent, but 20% opposed this. Preference for clinical/identifiable sharing decreased with geographical distance and was slightly less for MH than PH data, with small framing effects. Preference for research/de-identified data sharing without explicit consent showed the same small PH/MH and framing effects, plus greater preference for sharing structured data than de-identified free text. There was net support for research sharing to the NHS, academic institutions, and national research charities, net ambivalence about sharing to profit-making companies researching treatments, and net opposition to sharing to other companies (similar to sharing publicly). De-identified linkage to non-health data was generally supported, except to data held by private companies. We report demographic influences on preference. A majority (89%) supported a single NHS mechanism to choose uses of their data. Support for data sharing increased during COVID-19.

**Conclusions** Support for healthcare data sharing for direct care without explicit consent is broad but not universal. There is net support for the sharing of de-identified data for research to the NHS, academia, and the charitable sector, but not the commercial sector. A single national NHS-hosted system for patients to control the use of their NHS data for clinical purposes and for research would have broad support.

**Trial registration number** ISRCTN37444142.

### Strengths and limitations of this study

► Patient and public involvement in study design.
► Detailed questions measuring public opinion on health data sharing for clinical and research purposes.
► Large national sample giving high power, with quantitative analysis, sensitivity analyses to approximate known population demographics, and serendipitous examination of pandemic effects.
► Embedded randomised experiment to control and measure variation due to framing.
► The sample remained under-representative of some demographic groups despite weighting, with potential for unmeasured selection (including self-selection) bias reducing generalisability.

## INTRODUCTION

In the UK, health-related information is recorded routinely by healthcare professionals and patients within the National Health Service (NHS; England, Scotland, Wales) or Health and Social Care (HSC; Northern Ireland), henceforth 'NHS' for brevity. When combined with personal identifiers such as names and addresses, the data represent 'confidential patient information' (CPI),[1] used to provide care and managed according to standard principles.[2–7] It is 'owned' legally and managed by the NHS organisation recording it.[5 8] De-identified or anonymised forms of the data may be used for research (figure 1) without explicit consent,[5 6] as pledged by the NHS.[9] Identifiable data may be used for research with consent, or—under restricted circumstances—without.[1 5 6] 'Fully' anonymised data are not subject to UK data protection legislation.[5 6] However, even supposedly anonymised data relating to individual people carries some risk of reidentification via 'jigsaw' attacks.[10]

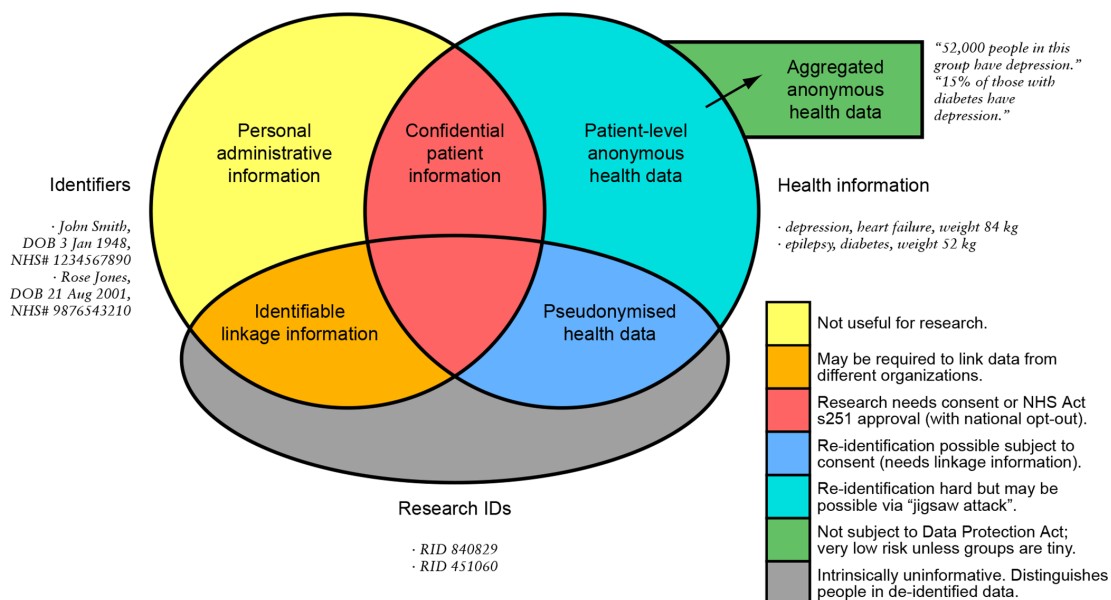

**Figure 1** Classifying health data according to identifiability. At the 'patient' level, the Venn diagram shows the overlaps between information that directly identifies a person, research identifiers (RIDs or pseudonyms), and health information, with simple examples. Anonymous health data may also exist in aggregated form, distinct from patient-level data; this aggregated form is the norm for public distribution. The level of identification risk and the research information governance requirements vary with the resulting categories of data. All examples are fictional. DOB, date of birth; NHS, National Health Service.

Our understanding of public wishes about data sharing is incomplete.[11] Information is sometimes not shared clinically when it should be,[3 12] and patients may be surprised and frustrated by failures to share in a 'national' health service.[13 14] Previous work, while establishing themes in public views on data linkage and sharing for research,[11 15–21] has highlighted the very small scale of many studies, and the paucity of research about the views of minority groups and the acceptability of sharing some types of data, such as mental health (MH) data. Mental illnesses can carry significant stigma[22] but are associated with substantial loss of life expectancy,[23] necessitating improvements in research and care. Some research requires multi-source data, but linkage is complex and may involve transient use of identifiable information.[24] It is unclear to what extent the public supports such work, and whether support varies with the type of data to which health data are linked (e.g., education vs criminal justice); there is little prior research in this area.[18] Proposed national systems for NHS data research such as 'care.data' have previously aroused public opposition and ire,[25] as have information governance (IG) breaches,[26] and there is current debate about the newest NHS data sharing proposal, General Practice Data for Planning and Research (GPDPR).[8 27]

What do patients and the public want now? We studied views on the sharing of identifiable health data (for clinical purposes) and de-identified health data (for research) within the UK. We examined data destinations ranging from local NHS services to public distribution. We distinguished MH and physical health (PH) data, and for research also structured data (coded information) versus free-text data (e.g., narrative information typed by clinicians).[28] We asked about data linkage for research. We examined the effects of decision 'framing', a term describing a decision-maker's conception of the possibilities and contingencies involved in a choice: since choices can be affected materially by the way in which they are formulated and presented,[29] we superimposed a randomised experiment to quantify how opinions on sharing were affected by the framing of risk versus benefit. We examined the effect of the COVID-19 pandemic on preferences. We sought views on potential systems to govern NHS data sharing for clinical and research purposes and to offer direct participation in research.

## METHODS

### Patient and public involvement

The research team advertised and formed a research advisory group (RAG) comprising patients and carers, who designed and co-produced the study and questionnaire with the research team (see online supplemental methods, S1.1). Patients, carers, and other members of the public participated in the study. Some members of the RAG co-wrote this paper.

### Inclusion criteria; sample size

The inclusion criteria were current residence in the UK and informed consent. The ability to take an online survey (alone or supported) was implicit. Participants under 16 years required the permission of their parent or guardian to participate and were asked to report whether they had assistance. We sought a power of 0·9 to detect a 'small' effect (Cohen's d=0·1) for the framing intervention (described below), with an estimated minimum

n=433/group, but beyond that sought a large sample of the UK population.

## Recruitment

Approvals covered public announcements and recruitment via health service sites, in person or through a variety of media. The study was adopted onto the National Institute for Health Research (NIHR) Clinical Research Network (CRN) portfolio; 216 general practice (GP) surgeries and 154 large healthcare organisations (e.g., acute care Trusts, MH Trusts, community hospitals, ambulance Trusts) supported recruitment. The study ran from 7 February 2020 to 30 September 2020.

## Survey

Data were collected using REDCap.[30] The survey is reproduced in the online supplemental methods, S1.2. It asked for the respondent's views on current and desirable practice for sharing identifiable data for clinical care purposes; personal experience of MH/PH conditions and care; preference for sharing identifiable PH/MH data (for clinical care purposes) to a range of NHS 'destinations'; preference for sharing de-identified structured PH/MH data (for research) to a range of potential research 'destinations'; similarly for data including de-identified free-text notes; views about potential systems for managing data consent in the NHS; views about linkage for research to non-NHS data sources; and demographics.

## Randomised framing intervention

We hypothesised that the context of questioning would affect willingness to share MH versus PH data, and sought to control and measure this effect. Before we asked about willingness to share different kinds of health data, we presented one of three framing statements: neutral, 'concern' (about MH data being more sensitive) or 'holistic' (about the importance of joined-up PH/MH care) (online supplemental methods, S1.2). Participants were randomised to one of the three statements.

## Data processing

Where participants agreed to leave a postcode, this was converted to a larger Office for National Statistics (ONS) geographical area, to prevent inadvertent identification. The geographical area was linked to its known population and Index of Multiple Deprivation (IMD). If the participant provided sufficient information, the ONS National Statistics Socio-Economic Classification (NS-SEC) was also calculated (See online supplemental methods, S1.3).

## Pandemic

By chance, our study spanned the UK onset of the COVID-19 pandemic. This had many consequences, including 'lockdowns'. Major changes were made to NHS data handling, including instructions to share CPI for public health purposes relating to the pandemic,[31] media reports of sharing of patient-level de-identified data with industry,[32] and guidance for GPs to include additional information in patients' Summary Care Record (SCR,

England) unless they had previously opted out.[33] We examined whether the pandemic was associated with changes in preference relating to data sharing, using 23 March 2020 (first UK 'lockdown') as the split point (factor 'pandemic': levels 'before lockdown', 'at/after lockdown').

## Analysis

We analysed using R v3.6.3.[34] We analysed categorical associations via $\chi^2$ tests, and effects on ordinal Likert-type scales (phrased linguistically to approximate interval scales) via analysis of variance (ANOVA). With a large sample size, the central limit theorem means that the distribution of means and mean differences tends to normal even though the parent population is non-normal, and ANOVA is robust to non-normality,[35–37] permitting ANOVA of discrete dependent variables. Scales measuring likelihood were quantified as −2 very unlikely, −1 unlikely, 0 not sure, +1 likely, +2 very likely. Yes/no scales were quantified as −1 no, 0 not sure, +1 yes. Models involving within-subject terms were analysed using the lmer and lmerTest packages, using type III sums of squares, and are expressed thus (~, 'is predicted by'; A×B, interaction; A*B denotes the inclusion of main effects A and B and their interactions). Statistics are shown to three significant figures (or as integers for percentages reported as annotations on figures or in the abstract/discussion) and degrees of freedom (df) are rounded to integers. We set α=0·05, and report 'NS' for 'not significant' and 'VLP' for a very low p (VLP) value, $p < 2 \cdot 2 \times 10^{-16}$.

Opinions on sharing clinical/identifiable data were analysed using a model termed C1: *sharing~destination\*nature\*framing\*pandemic+(1|subject)*. 'Destination' had four levels (local, regional, national, UK-wide), 'nature' had two (PH, MH), and framing had three (neutral, MH concern, holistic). We followed up nature×framing interactions by analysing MH and PH data separately using the simplified model C1B: *sharing~destination\*framing+(1|subject)*.

To examine the effects of demographic factors and experience, we used a larger model, C2: *sharing~destination\*nature\*framing\*pandemic+age+gender+ethnicity+education+sexuality+religion+nation+imd_quartile+nssec+mh_experience\*nature+(1|subject)*. This was only possible for people who provided all necessary demographic information. Levels for demographic factors were as per online supplemental table 1, plus sexuality (two levels: heterosexual/straight, LGBT+ (including homosexual/gay/lesbian, bisexual, other/self-described)) and NS-SEC (five levels). We did not include all interaction terms for demographic factors (as formal tests of 'intersectionality' effects) because of the combinatorial explosion this would entail; instead, this model tests main effects of demographic factors plus the specific hypothesis that MH experience affects sharing of MH/PH data differentially.

Opinions on sharing de-identified data for research were analysed using model R1: *sharing~destination\*nature\*detail\*framing\*pandemic+(1|subject)*. 'Destination' had six levels (NHS, academia, charities, companies conducting

treatment research, other companies, publicly); 'detail' had two levels (structured only, free text); other factors were as before. To examine nature×framing interactions, we used the simplified model R1B: *sharing~destination\*-framing+(1|subject)*. For demographic analysis, we used model R2: *sharing~destination\*nature\*detail\*framing\*pandemic+age+gender+ethnicity+education+sexuality+religion+nation+imd_quartile+nssec+mh_experience\*nature+(1|subject)*.

Sensitivity analyses were conducted by weighting to UK population demographic proportions. Effect size plots were created for key models. (See online supplemental methods, S1.4–S1.5.)

Willingness for linkage to non-NHS data for research (data source, eight levels) was analysed for all participants using model L1: *willingness~source\*pandemic+(1|subject)*. For demographic analysis, we used model L2: *willingness~source\*pandemic+age+gender+ethnicity+education+sexuality+religion+nation+imd_quartile+nssec+mh_experience+(1|subject)*.

A thematic analysis was performed on free-text comments (see online supplemental methods, S1.6).

## RESULTS

### Participants

Consenting participants numbered 29 275. Recruitment is shown in online supplemental figure 1A,B; 8019 participated before UK 'lockdown' and 21 256 on/after that date. Not everyone completed the survey: participation by stage is shown in online supplemental figure 1C, with 73·6% completing all stages. Median completion time was 18·4 min. Participants were evenly distributed across framing conditions (neutral 9812, MH concern 9744, holistic 9719; $\chi^2_2$=0.475, NS).

Demographics are shown in online supplemental figure 2A–J (with free-text responses in online supplemental results, S2.2). Relative to the UK population (online supplemental table 1), our sample under-represented the youngest and oldest age ranges, males, those of non-white ethnicity, those with less formal education, those professing a religion, residents of UK nations other than England, and people living in more deprived areas. Weighting yielded substantial though incomplete improvement (online supplemental results, S2.4). There was coverage of most UK local authority areas (online supplemental figure 2I).

An MH condition had been experienced by 40·0% of participants (online supplemental figure 3), primarily depression and anxiety disorders (of people who had experienced an MH condition, 93·8% reported having had depression or anxiety at some point). Of participants who had experienced an MH condition, 84·9% had used MH services, primarily their GP and NHS psychological therapy services (online supplemental figure 3). PH services had been used by 88·2% of respondents, primarily GP and outpatient services (online supplemental figure 3).

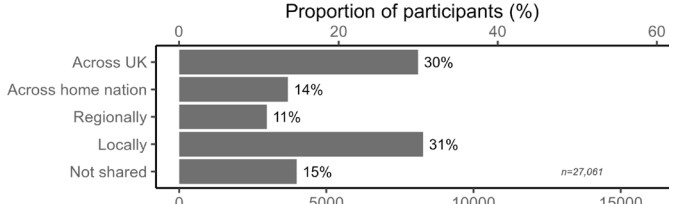

**A. Understanding of current clinical sharing**

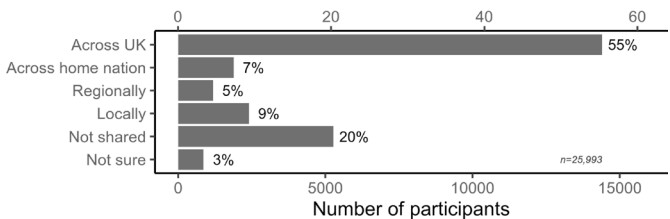

**B. Preference re clinical sharing without consent each time**

**Figure 2** (A) Understanding of how health data are shared identifiably without explicit consent for clinical purposes, and (B) preference as to what should happen. The denominator for percentages is the number of people who answered each question, shown at the bottom right of each panel.

### Sequence

We first report preferences for clinical data sharing based on multiple-choice questions (figure 2), before examining in detail participants' likelihood of sharing data for clinical and research purposes (figure 3, with weighted equivalent in online supplemental figure 4), preferences for data linkage (figure 4, online supplemental figure 5), and changes associated with the pandemic (figure 5). We then report effect sizes for the analytical models, including for demographic factors (figure 6A–C, online supplemental figure 6A–C), and finally report views on a possible national consent system (figure 7A–F, online supplemental figure 7A–F).

### Sharing identifiable data for clinical purposes

Understanding of current NHS practice regarding identifiable data sharing between care providers, without asking the patient each time, is shown in figure 2A. In practice, sharing varies by area, for example depending on whether a local/regional shared care record (ShCR) is operative in part of England,[14] or according to limited national systems such as the Intra-NHS Scotland Information Sharing Accord[38] and Scottish Emergency Care Summary,[39] the Northern Ireland Electronic Care Record,[40] the English SCR,[41] and a variety of systems in Wales.[42] To our knowledge, there is no UK-wide sharing, but 30·0% of respondents thought that there was free sharing of identifiable data across the UK.

When asked preferences via a single multiple-choice question (figure 2B), there was majority (55·4%) support for sharing identifiable data for direct care across the UK, without being asked first, and 76·4% supported sharing at least locally, but a substantial minority (20·3%) said that sharing should not occur without the patient being asked first.

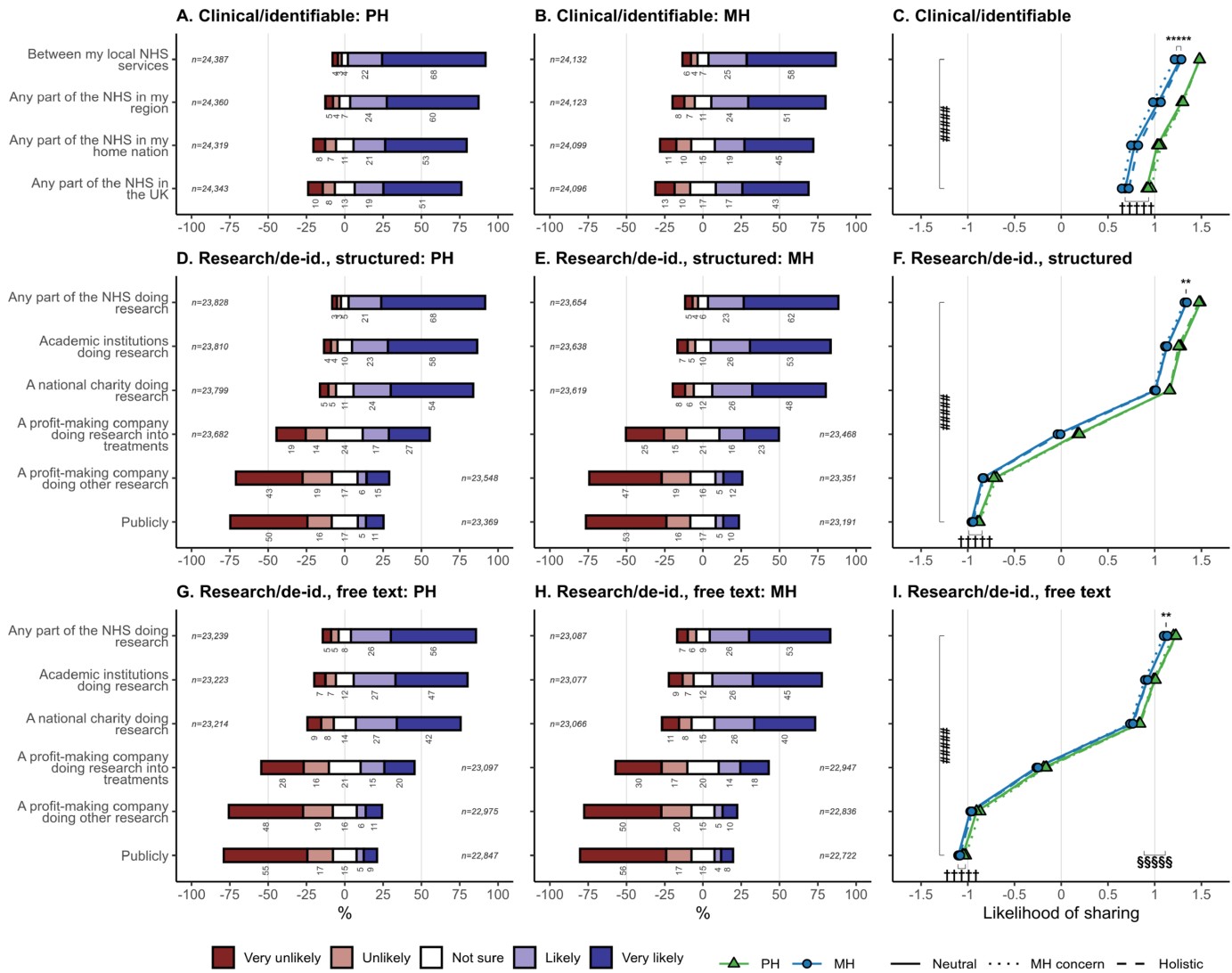

**Figure 3** Participants' self-reported likelihood of sharing mental and physical health data without explicit consent for clinical purposes (identifiably) or research (de-identified), according to destination, nature (MH vs PH), framing statement, and for research purposes also the level of detail (structured only vs with free text). The denominator for percentages is the number of people who answered each question. In (C), (F) and (H), the abscissa is the mean of responses coded as −2 very unlikely, −1 unlikely, 0 not sure, +1 likely, +2 very likely. Analyses were from models C1 and R1 as described in the Methods. See online supplemental figure 4 for corresponding weighted analysis. De-id, de-identified; MH, mental health; NHS, National Health Service/Health and Social Care; PH, physical health; ####p<10$^{-5}$ for main effect of destination; †††††p<10$^{-5}$ for main effect of nature, with bar length showing mean difference between MH and PH; **p<0.01 and *****p<10$^{-5}$ for framing×nature interaction, with bar showing the mean difference between 'MH concern' and 'neutral'; §§§§p<10$^{-5}$, main effect of detail, comparing (F) with (I), with bar length showing the mean difference between structured and free-text conditions.

## Sharing MH and PH data for clinical purposes and for research

Willingness to share health data without being asked every time is shown in figure 3 by purpose, nature, and destination (with corresponding weighted data in online supplemental figure 4).

For clinical purposes (with identifiable data), there was strong net willingness to share (figure 3A–C), with 89·9% (PH data) or 83·1% (MH data) rating themselves 'likely' or 'very likely' to share to local NHS services. The most important determinant was destination, with stronger support the more local the sharing (i.e, preference decreased with geographical distance). This monotonic effect, when participants were asked to rate

each destination and nature separately, had not been evident in the one-from-many question about health data in general, asked previously (figure 2B). People were slightly more willing to share PH than MH data. There were significant but very small effects of the framing statement, primarily that 'MH concern' framing reduced willingness to share MH data. In the whole-sample analysis (model C1), there were highly significant effects of destination ($F_{3,169348}$=6490, VLP) and nature ($F_{1,169484}$=6080, VLP), as well as interactions including nature×framing ($F_{2,169484}$=78·4, VLP) (figure 3C, online supplemental figure 6A). This interaction was driven primarily by a simple effect of 'MH concern' framing to reduce sharing

**De-identified linkage to non-health data**

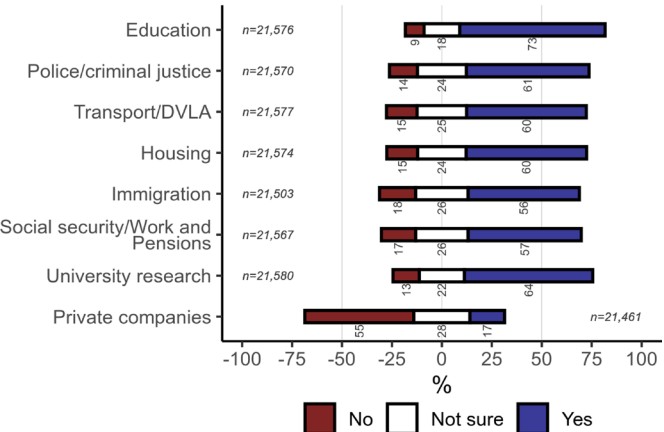

**Figure 4** Participants' willingness for their National Health Service/Health and Social Care (NHS) data to be linked to non-health data of different kinds for research. The denominator for percentages is the number of people who answered each question. See online supplemental figure 5 for corresponding weighted analysis. DVLA, Driver and Vehicle Licensing Agency.

for MH data (model C1B: PH data, no effect of framing ($F_{2,24461}=1.18$, NS); MH data, effect of framing ($F_{2,24157}=8.36$, p=0·000234); pairwise comparison within MH data, MH concern vs neutral, p=0·00443). Framing effects were also

lessened for geographically broader destinations. Effect sizes are reported further below.

For research purposes (with de-identified data), destination was an extremely strong driver of preference ($F_{5,535334}=87\,800$, VLP) (figure 3D–I). On average, people expressed strong support for sharing to the NHS, academia, or national charities for research purposes. For NHS sharing, the most popular destination, 89·2% (PH data, structured only), 85·2% (MH data, structured), 81·6% (PH data, free text), and 78·7% (MH data, free text) of participants rated themselves 'likely' or 'very likely' to share. Support and opposition were approximately equally balanced for sharing to profit-making companies researching treatments. There was strong net opposition to sharing to other types of companies, approximately equal to that for sharing publicly. Only 20·6% (PH data, structured only), 17·6% (MH data, structured), 16·4% (PH data, free text), and 14·9% (MH data, free text) rated themselves 'likely' or 'very likely' to share to other types of companies. There was a small but significant preference for sharing PH (vs MH) data, and likewise higher preference for sharing structured-only versus free-text data. In the whole-sample analysis (model R1), there were highly significant effects of destination, nature, and detail, plus interactions including destination×nature×detail (online supplemental figure 6B).

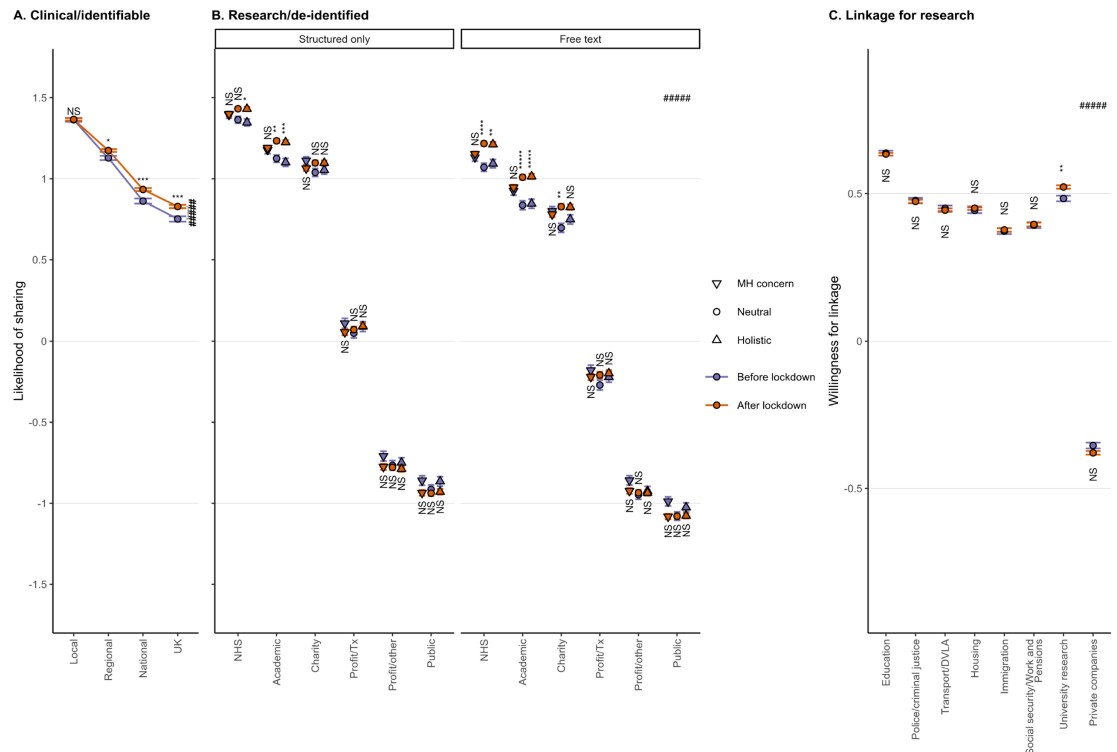

**Figure 5** Change in preference in relation to the COVID-19 pandemic. (A) Clinical/identifiable data sharing, by destination. Dependent variable (preference for sharing) as for figure 3C, now shown on the ordinate (y) axis. (B) Research/de-identified data sharing, by detail (structured vs free text) and destination. Dependent variable as for figure 3F,H. (C) Linkage for research, by non-NHS data source type. Dependent variable coded as –1 no, 0 not sure, +1 yes. DVLA, Driver and Vehicle Licensing Agency; MH, mental health; NHS, National Health Service/Health and Social Care; NS, not significant. Error bars show ±1 SEM; *****$p<10^{-5}$, ****$p<10^{-4}$, ***$p<10^{-3}$, **$p<10^{-2}$, *$p<0.05$ by two-sample t test Šidák-corrected for multiple comparisons; ####$p\lll 10^{-5}$, destination×pandemic interaction.

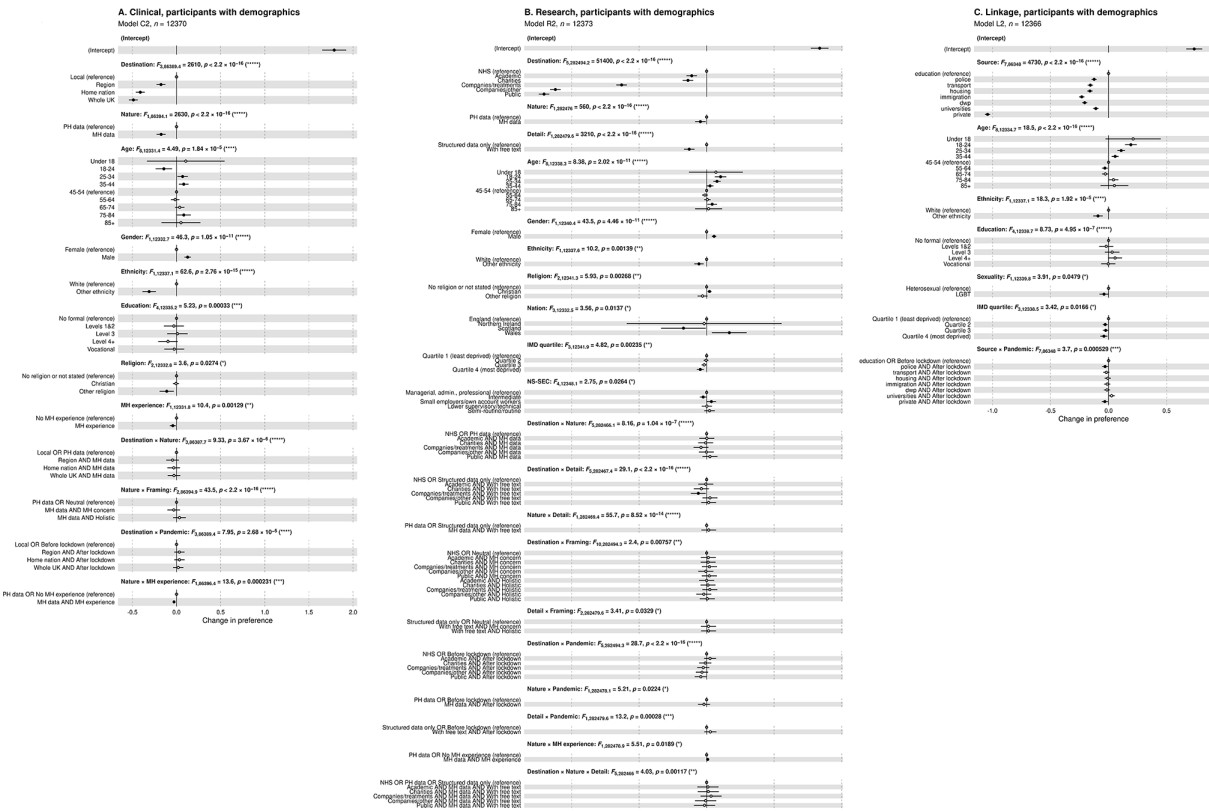

**Figure 6** Effect sizes for (A) clinical data sharing via statistical model C2, (B) research data sharing via model R2, and (C) linkage via model L2. These models include only those participants who supplied full demographic information, to allow analysis by demographics; compare online supplemental figure 6 (all participants). Only those model terms with a significant *F* test are shown. Effect sizes with 95% confidence intervals are shown for each level as uncorrected pairwise comparisons to a reference category within each term (note the difference in what is being tested pairwise vs the omnibus *F* test for the term; see online supplemental methods). ●p<α; ○ NS. MH, mental health; NHS, National Health Service/Health and Social Care; PH, physical health.

Framing effects included nature×framing, though simple framing effects were not significant for PH or MH data separately (model R1B). Effect sizes are reported further below.

Sensitivity analyses weighted to UK population demographics (online supplemental results, S2.5) were consistent with the primary analysis.

### Linkage to non-health data for research
We asked about linking of NHS data to non-health data sources for research, ultimately with de-identified data. There was net support for all 'state' sources and university-held data (figure 4), ranging from 72·8% support for education data to 56·6% support for social security/work and pensions data, but net opposition regarding private company data (figure 4), for which only 17·3% were supportive. Weighted responses were very similar (online supplemental figure 5).

### Changes related to the COVID-19 pandemic
Following 'lockdown', willingness to share identifiable data for clinical purposes increased, with no significant change in the already high preference for local sharing, but progressive increases for sharing to more remote

parts of the NHS (model C1, destination×pandemic, $F_{3,169348}$=26·6, VLP; figure 5A).

Willingness to share de-identified data for research purposes generally increased for more-preferred destinations (NHS, academia, charities), except in the 'MH concern' framing condition (model R1, destination×pandemic, $F_{5,535334}$=78·2, VLP; figure 5B), but did not change for less-preferred destinations (commercial and public sharing).

Preference for linkage to university data increased (source×pandemic; figure 5C; online supplemental figure 6C). There was a less consistent decrease in preference for linkage to private data (figure 6C, online supplemental figure 6C) and police data (model L2; figure 6C).

### Effect sizes and influence of demographic factors
Preference varied according to demographic factors and experience of MH illness. For clinical purposes, there were several demographic effects (model C2; figure 6A shows significant terms with effect sizes). Age was a significant factor, with the age bands most willing to share being 25–44 and 75+, and the 18–24 band being least willing. Males were more willing to share data than females.

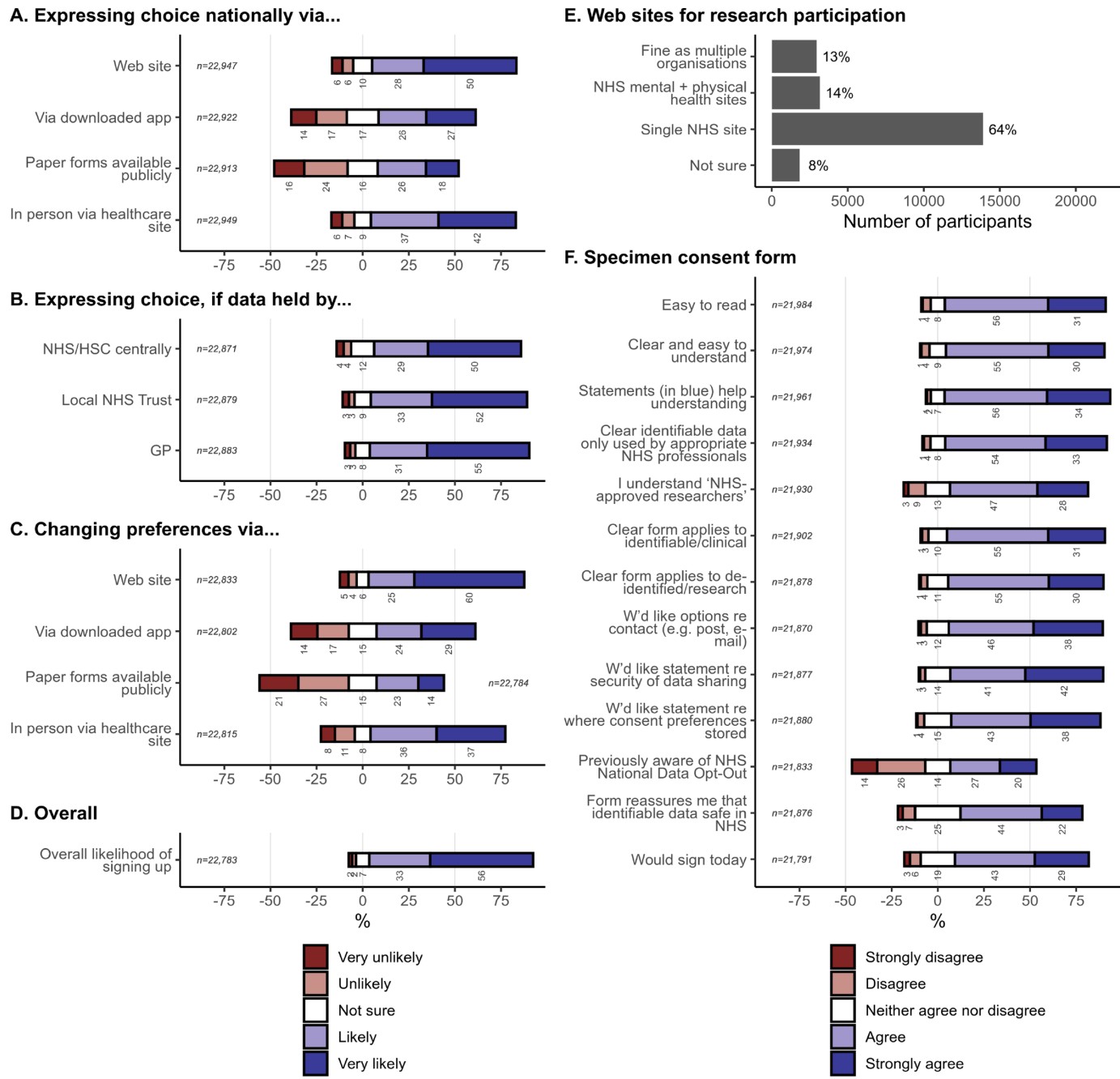

**Figure 7** Views on a national data sharing consent system. The denominator for percentages is the number of people who answered each question. See online supplemental figure 7 for corresponding weighted analysis. GP, general practice; HSC, Health and Social Care; NHS, National Health Service.

Those of minority ethnicity were less willing to share than those of white ethnicity. There was a main effect of education, and across educational levels, those of Level 3 were most willing and those of Level 4+ least willing. Those of minority religions were less willing to share. Those from the most-deprived IMD quartile were also less willing. There were no significant effects of sexuality ($F_{1,12334}$=2·21, NS), NS-SEC ($F_{4,12335}$=1·32, NS) or nation ($F_{3,12335}$=2·13, NS). Personal experience of MH illness specifically reduced willingness to share MH data for clinical purposes (nature×MH experience).

For research purposes, significant effects were similarly observed for age, gender, ethnicity, religion, and IMD quartile (model R2, figure 6B). The age distribution was clearly U-shaped, with greater willingness to share among the youngest and oldest groups. As before, there was no effect of sexuality ($F_{1,12336}$=1·58, NS). There was no effect of education ($F_{4,12348}$=2·15, p=0·072), but there was an effect of nation (with people living in Wales more willing to share and those in Scotland less so, relative to England), and of socioeconomic status (NS-SEC; figure 6B). People with MH experience were significantly more likely to

share MH data for research purposes (nature×MH experience, $F_{1,296111}$=6·15, p=0·0132).

For linkage, the patterns were broadly as before (figure 6C, online supplemental figure 6C). Data source strongly influenced preference (education>universities≈police≈housing≈transport>social security>immigration≫private companies). There were also effects as before of age, ethnicity, education, and IMD quartile. There was no effect of gender ($F_{1,12338}$=2·32, NS), religion ($F_{2,12338}$=1·86, NS), nation ($F$<1), or MH experience ($F$<1), but there was now an effect of sexuality, with LGBT+ people being less willing for linkage.

## A possible national consent system

We proposed varieties of a national system for patients to decide how their NHS data are used. Participants were most willing to sign up via a website or in person (figure 7A and C). Willingness was similar regardless of whether consent information was managed by the NHS centrally, a local NHS Trust, or the patient's GP (figure 7B). Overall, 88·8% of people said they were 'likely' or 'very likely' to sign up to such a system (figure 7D). Weighted responses were very similar (online supplemental figure 7).

Most people (63·6%) wanted a single NHS system to sign up for participatory research (figure 7E; online supplemental figure 7E).

There was broad support for the draft consent form and for adding information about contact methods, data security, and management of the consent information, with 66·1%–89·8% answering 'agree' or 'strongly agree' to all questions except about awareness of the NHS National Data Opt-Out (NDO) (figure 7F; online supplemental figure 7F); 46·7% (unweighted) agreed that they had been aware of the NDO.

Common subthemes from the thematic analysis (n>100, online supplemental results, S2.9–S2.10) included: the need for clarity around de-identification (n=164); the critical importance of healthcare data security (n=594); the desirability of data sharing (n=290); that opt-outs should be more prominent (or default) or linked to the NDO (n=134); that profit-making use should not happen or that the NHS/patients should benefit from such profits (n=198); that clinical users should be specified in more detail (n=147); research users likewise (n=268); and that healthcare data should not be available to private or third-party companies without specific permission (n=417). For full theme/subtheme descriptions and tallies, see online supplemental results, S2.9.

## DISCUSSION

### Summary

Many respondents believed that health data are shared UK-wide for clinical purposes without explicit consent, when sharing is usually more limited. A majority (76%) supported such sharing, though a significant minority (20%) opposed it. Geographically broad sharing was endorsed, though with stronger support for more local destinations. People preferred to share PH (vs MH) data, but this was less important than the destination.

For research, with de-identified data, there was strong net support for sharing without explicit consent to the NHS, academic research institutions, and research charities. There was net ambivalence regarding private companies researching treatments, and strong net opposition for sharing to other companies or publicly. There was a small preference for sharing PH over MH data (a smaller difference than for clinical purposes), and greater support for structured-only data over de-identified free text. There was net support for research linkage to state and university data sources, but opposition regarding data held by private companies.

Framing statements influenced MH/PH preferences, but only to a small degree. Age, gender, ethnicity, education, religion, and IMD were associated with willingness to have health data shared or linked, though not nearly as strongly as destination/source. Personal experience of MH conditions was associated with reduced willingness to share MH data for clinical purposes, but greater willingness to share it for research. After COVID-19 lockdown, there was greater willingness regarding already-preferred destinations.

Respondents endorsed a suggested UK-wide system allowing patients to control the clinical/research uses of their data and to sign up for participatory research, with 89% saying they would be likely or very likely to use such a system. In commenting, respondents frequently emphasised the importance of data security and that NHS data should not be made available to private or third-party companies without specific permission. Comment themes reflected the tensions previously noted in this area, including about healthcare and research benefits, security, governance, trust, and vulnerable groups.[43–45]

### Strengths and weaknesses

Strengths include patient/public involvement in the study design; the largest such study to date by 1–3 orders of magnitude,[15 17 19–21 46] giving high power; sensitivity analyses weighted to population demographics; detailed questions about data sharing for clinical/research purposes, including about the type of data and the destination, plus linkage to non-health data; a randomised framing experiment to control and measure this source of variation; quantitative analysis including of relative effect sizes; consultation on ways to improve the current situation; and serendipitous examination of the effects of COVID-19 on data sharing views.

The major weakness is that the sample remained under-representative of some groups despite weighting, with potential for unmeasured selection (including self-selection) bias, reducing generalisability. We consider the reasons for the bias, and potential routes to improving representativeness in future work, in the online supplemental discussion, S3.1.

### Destination and purpose

The Caldicott framework (1997)[47] and reviews (2013, 2016),[3 12] regarding safe information sharing for direct clinical care, included the principle that the 'duty to share information can be as important as the duty to protect patient confidentiality'[3] and noted that information was often not shared when it should be, for fear of inappropriate disclosure.[12] That was despite legislation creating a duty on providers to share information with professionals when that is likely to facilitate the individual's health or social care, disclosure is in their best interests, and they do not or are not likely to object.[48] That is in essence an opt-out system. This legislation conflicted with some prior studies of public opinion.[49] We provide more background on relevant legislation in the online supplemental discussion, S3.2. The 2016 Caldicott review noted low public understanding around how health information is used, but 'an expectation that information is shared for direct care'.[12] We observed net support for such sharing that varied with geographical destination and was by no means universal, but was nevertheless strong.

In relation to research and other non-clinical activities, the recommendation that people be able to opt out from personal confidential data being used beyond their own direct care[12] led to the NHS NDO.[50 51] This relates to the use of CPI (identifiable information) for purposes such as research, conducted under NHS Act Section 251 (§251) approvals.[1 52] It does not apply to direct clinical care, local audit or service evaluation, or de-identified information.[50 51] Our study and others show it remains unknown to many.[46] Furthermore, it is not simple (see online supplemental discussion, S3.2) and we suspect many do not fully understand its scope. Conversely, from the researcher's perspective, §251 approval is often still required for linkage studies in which researchers never see identifiable information: there is no standardised 'trusted third-party' system for centralised linkage of identifiable information, and inconsistent adoption of de-identified linkage methods.

'Destination' was by far the strongest driver of preference for sharing and linkage. This pattern is established: willingness to allow researchers/clinicians access to health data, but far greater reservations about industry.[53 54] An important basis for this is mistrust of the security and/or motives of commercial organisations,[25 26 55] as our participants noted.

### Demographic effects

A common demographic theme was that minority groups (of ethnicity, religion, and sexuality) and deprived groups were less willing to share. This might reflect experience of disadvantage to, or discrimination against, these groups.[56 57] Ethnicity has had mixed effects on preference for national electronic health record (EHR) systems.[19] In our study, age effects were generally biphasic, with higher willingness amount the youngest and eldest. Youth may be associated with familiarity with data and/or greater support for EHRs,[58] and older age

with an increasing burden of illness, itself associated with support for national EHR systems.[58] Educational effects were relatively inconsistent, being small and present for clinical and linkage preferences though in quantitatively different ways, and not being significant for research preferences. Males were slightly more willing to share than females. Similar results have been observed before,[46] but not always;[58 59] one reason might be gender-based healthcare discrimination.[60] Higher support for research sharing in Wales may relate to established national research systems there;[61 62] the reasons for reduced willingness in Scotland are unclear, but similar systems there are younger.[63] Those with personal experience of MH illness were less willing to share identifiable MH data for direct care. This may reflect experience of discrimination or stigma[64 65]—which can have disproportionate effects in subgroups.[65 66] However, the same people were more willing to share de-identified MH data for research, potentially reflecting increased prioritisation of MH research.[67]

Demographic variations in preference may reflect differences in perception of current data rules or security practices, reasons for concern about uses of health data, or degree of concern. UK law prohibits variation of policy according to these factors.[68] Better understanding and public information may be required to address these groups' concerns,[11 69] but improvements in health equity are also required.[17 56] However, the effect sizes of these demographic predictors were not large enough to override the net support for data sharing, given the right destinations.

### Framing and pandemic effects

We observed small but significant framing effects.[29] Our framing statements were true and non-alarmist, so real-world framing effects might sometimes be larger. Others have observed larger effects via 'loss framing' (emphasising the potential adverse consequences of not consenting over the potential benefits of consenting), and through other manipulations such as the placement of framing statements.[70] Media coverage of health data sharing is influential.[71 72] Despite best intentions, it is impossible to avoid framing effects entirely,[29] so those presenting information should be aware of these while presenting accurately the risks and benefits of data sharing/linkage.

During COVID-19, despite press coverage[32] of an enforced increase in NHS data sharing for public health purposes,[31 33] support for sharing/linkage increased—but only for some already-favoured destinations. Publicity regarding NHS care[73] and research regarding COVID-19[74 75] may have driven the increase in support for sharing with the NHS, universities, and research charities. We did not analyse the pandemic-related trajectory of responses beyond examining the change at/after the first UK 'lockdown', and public views may have changed further after the conclusion of our study.

## CONCLUSIONS

Participants supported a central system for patients to control the uses of their data, and likewise a single NHS mechanism to sign up for active research participation. There is a trade-off between the scientific desirability of everyone contributing de-identified data, including to avoid bias,[76 77] and the desirability of individual control over data use.[78] As we suggest below, a reasonable balance might be a central system to opt out from identifiable clinical use, identifiable (§251) research use, or de-identified research use of one's data, and to opt in for participatory research. This would complement efforts to improve people's access to their own data.[79]

The majority support that we observed for clinical sharing without explicit consent perhaps makes such sharing reasonable as a default (opt-out) position, given the potential advantages for many people's own care, subject to strict IG rules (who has access, and when). Under UK data protection laws, the legal basis for NHS organisations to hold patient data is not consent (see online supplemental discussion, S3.2). The Caldicott principles require relevant information sharing for direct care under many circumstances,[3 12] and NHSX (NHS User Experience) have set out the IG requirements for regional ShCRs, several of which are already in operation (and may use opt-outs), and for cross-ShCR sharing.[80] However, a significant minority of participants in our study opposed clinical sharing without explicit consent, mandating (we suggest) at least a public information campaign about opt-outs, sufficiently targeted to reach groups most concerned about data sharing, if broader sharing were to occur, to conform to the Caldicott principle of 'no surprises'.[3]

There was strong net support for NHS, academic, and charity researchers accessing de-identified health data. Opt-outs are often offered, even if they are not legally required, for research using de-identified data (see online supplemental discussion, S3.2). A standard method for conducting such research is via a trusted research environment (TRE).[24 61 81–83] Approved researchers come 'into' the secure environment, which can be highly controlled, to interact with relevant data (e.g., pseudonymised; figure 1). After analysis, aggregation, and other statistical disclosure control (SDC),[84] results go 'out' for publication (figure 1). The principle is of the 'five safes': safe people, safe projects, a safe setting, safe data, and safe outputs.[83] The lower preference for sharing de-identified free-text data versus de-identified structured data is congruent with the increased sensitivity of free-text data, and the fact that technical methods for de-identifying free text remain imperfect;[85 86] additional safeguards surrounding that kind of data are justified.

We did not examine preferences regarding research uses of identifiable data, except about being contacted for direct research participation and about awareness of the NDO. The NDO only applies to work using identifiable data without consent (online supplemental discussion, S3.2). Separately, some argue that it should be easier to gain permissions to conduct research using identifiable or potentially identifiable data, as part of an ethical duty to participate in research.[87] We provide no evidence to suggest changes to the operation of this process, but we found strong support for a central mechanism to control the uses of one's data and to sign up for direct research participation.

In contrast to the support for NHS/academic/charity research, respondents did not support research sharing to private companies. Some have suggested this is addressable in part by public education.[11] We suggest respecting public preference, and not giving commercial organisations direct access to patient-level NHS data for research, even de-identified, without consent. (This is distinct from the common NHS practice of employing companies, such as EHR software providers, to manage NHS data securely for clinical purposes.) We think that this does not rule out all industrial research uses of data, which could happen according to at least three methods. The first is via consent, as for commercial treatment trials. Second, companies could collaborate with NHS/academic researchers. For example, an artificial intelligence company could provide an untrained algorithm; NHS staff could train it on patient-level data; the company could receive a trained algorithm back while never having access to the data (assuming verification that the algorithm cannot 'embed' detailed data features during training). Third, methods exist whereby software queries come 'in' to the TRE, and semi-automatic or automatic SDC occurs before results go 'out'.[88–90] This allows research to take place without researchers having access to patient-level data, and can also support 'federated' queries across sites. Data that have undergone suitable SDC (e.g., aggregation) can be published, and are therefore suitable for industrial access if desired. Regardless, as our participants commented, the NHS might charge for such access,[11 91] and full transparency is essential. Formal, consultation-based standards governing this NHS–commercial interface would be desirable.

Some noted the need to control the nature of information sharing in greater detail, and this would need further detailed consultation. Our data may help to frame this. There is likely to be a trade-off between the level of fine-grained control offered and a need for simplicity in a nationwide system; our effect sizes (figure 6B) suggest, for example, that 'destination', and whether free text or only structured data are involved, should be prioritised over other factors such as MH versus PH data.

Governance of UK health data must be transparent and reflect the views of patients.[11 16] Regardless of legal authority, it is important that health data are processed in ways that have a 'social licence'.[92] As the UK Government seeks to change data legislation[93] and emphasise health data in its science strategy,[94] we hope this study contributes to the conversation.

**Acknowledgements** We thank all those who took part in the study and gave their views. We thank all the investigators, local collaborators, local NIHR CRN

staff, and healthcare staff across the UK who supported the study. We thank also Elizabeth Rotherham (research advisory group member) for help with design; Nicola Gleadle for practical support; Jane Gaffa, Rachel Kyd, and Mary-Beth Sherwood for Research and Development support; Sally-Anne Hurford and Ruth Hudson for NIHR CRN advice; Chess Denman for investigator approval; Laura Marshall for CPFT publicity; Hannah Clarke for helpful comments; Research Ethics Committee and Health Research Authority staff for ethical and regulatory approvals; the Cambridge Autism Research Database for additional publicity; and Artur Arikainen, Andres Colubri, Lisa Eckstein, Julia Ivanova, and David Wyatt for helpful comments during open peer review.

**Contributors** RNC conceived the study. LAJ, JRN, PHA, MRG, MP, and RNC designed it (with Elizabeth Rotherham). LAJ coordinated its execution. JMF advised on REDCap implementation and execution. RNC, LAJ, and JMF had access to the original data; the final (anonymous) data set is public (with participants' consent). RNC analysed the data. RNC and LAJ drafted the manuscript. All authors edited the manuscript and approved submission. RNC is guarantor.

**Funding** Supported by a UK Medical Research Council (MRC) Mental Health Data Pathfinder award (MC_PC_17213 to RNC), and MRC grant MR/W014386/1 (Health Data Research UK Mental Health Data Hub, DATAMIND). Recruitment was supported by the NIHR CRN and the REDCap installation was supported in part by the NIHR Cambridge Biomedical Research Centre (BRC-1215-20014); the views expressed are those of the authors and not necessarily those of the NHS, the NIHR, or the Department of Health and Social Care.

**Competing interests** None declared.

**Patient and public involvement** Patients and/or the public were involved in the design, or conduct, or reporting, or dissemination plans of this research. Refer to the Methods section for further details.

**Patient consent for publication** Not applicable.

**Ethics approval** This study involves human participants and was approved by the NHS Research Ethics Service: East of Scotland Research Ethics Service, reference 19/ES/0144. The study was pre-registered (https://doi.org/10.1186/ISRCTN37444142). Participants gave informed consent to participate in the study before taking part.

**Provenance and peer review** Not commissioned; externally peer reviewed.

**Data availability statement** Data are available in a public, open access repository. After removal of all free text, anonymised data are available from the University of Cambridge Data Repository at https://doi.org/10.17863/CAM.75784, with participants' consent.

**ORCID iDs**
Linda A Jones http://orcid.org/0000-0001-9347-5715
Rudolf N Cardinal http://orcid.org/0000-0002-8751-5167

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
