## [Reviewer comments · BMJ Open]

ARTICLE DETAILS

TITLE (PROVISIONAL)	Public opinion on sharing data from health services for clinical and research purposes without explicit consent: an anonymous online survey in the UK
AUTHORS	Jones, Linda; Nelder, Jenny; Fryer, Joseph; Alsop, Philip; Geary, Michael; Prince, Mark; Cardinal, Rudolf

VERSION 1 – REVIEW

REVIEWER	Wyatt, David King's College London, Division of Health & Social Care
REVIEW RETURNED	26-Oct-2021

GENERAL COMMENTS	Thank you for inviting me to review this well written and fascinating paper, which draws on a vast amount of survey data to explore public opinion on data sharing for health and research purposes. Focusing on data sharing without consent, the authors document key new information in contemporary discussions of data sharing. Most notably, differences between physical and mental health data, age, gender, ethnicity, particularly in the context of sharing to the private sector. While existing research has highlighted this general distrust and resistance to routine health records being “sold” or utilised by the private sector, this study is the first I’ve seen that adds texture to our understanding of the contours of these attitudes. Such accounts are crucial if we are to explore ways of facilitating research while also addressing the needs and concerns of the population. I have provided comments below and some suggestions that I hope will help the authors extend their analysis and discussion (on what is already a very good and interesting paper). Methods: PPI involvement - It is clear from the author’s list and the Contribution description that RAG members were heavily involved in this project. However the statement on pg 3 doesn’t really do justice to this involvement - how was this arranged? How was the study designed in practice? How were the different expertise of the group managed to produce a socially robust but also academically sound survey? I’m pushing here because it’s listed as one of the
--

	strengths of the study and it's important to demonstrate HOW PPI has helped and improved this process. In terms of the quantitative analysis, I am not equipped to assess this meaningfully. The design decisions, however, and questions asked seem reasonable, particularly the inclusion different framings. Results: These are clearly presented in text. I struggled a bit with the references to so many different tables and supplementary material items, but this is more a reflection on journal submission formatting criteria than the manuscript itself! Discussion: Particularly in terms of the discussion (and possibly background), I think there is a lot of relevant work on data sharing for research in primary care (from a more social science perspective) which highlights some of these issues and takes it further to stress the need for trust, transparency and clear information governance. I think engaging with some of the themes in these papers (I've included references to a couple below), will help further develop the depth of the discussion. Some of the themes in the literature cited below also seem to resonate well with the free text answers (responding to the consent form) and discussed on Pg 8. GREENHALGH, T., WOOD, G. W., BRATAN, T., STRAMER, K. & HINDER, S. 2008. Patients' attitudes to the summary care record and HealthSpace: qualitative study. British Medical Journal, 336, 1290-1295. HADDOW, G., BRUCE, A., SATHANANDAM, S. & WYATT, J. C. 2011. Nothing is really safe': a focus group study on the processes of anonymising and sharing of health data for research purposes. Journal of Evaluation in Clinical Practice, 17, 1140-1146. STEVENSON, F. 2015. The use of electronic patient records for medical research: conflicts and contradictions. BMC Health Services Research, 15, 124. You may also want to look at Carter et al. (2015) to see how we might conceptualise the parameters of public acceptance of certain data sharing practices (discussed in the context of care.data but relevant beyond this). CARTER, P., LAURIE, G. T. & DIXON-WOODS, M. 2015. The social licence for research: why care. data ran into trouble. Journal of medical ethics, 41, 404-409. In relation to Trusted Research Environments, I can see how these would be an important way of circumventing data control. I think it's important to also acknowledge that the shift towards digital
--	--

	recording practices has meant that methods like TREs are key ways we can accurately control who has and does not have access to certain data sets (and how they are used). Outside of platforms like TREs, digital data makes these kinds of controls so much harder. A key reference for TREs not currently in the reference list is linked below: UK HEALTH DATA RESEARCH ALLIANCE 2020. Trusted Research Environments (TRE): A strategy to build public trust and meet changing health data science needs. https://ukhealthdata.org/wp-content/uploads/2020/04/200430-TRE-Green-Paper-v1.pdf I hope the authors find these comments helpful. Thank you once again for the opportunity to read this fascinating and important paper.
--	--

REVIEWER	Eckstein, Lisa University of Tasmania , Faculty of Law
REVIEW RETURNED	29-Oct-2021

GENERAL COMMENTS	This was a highly interesting dataset with nuanced information that no doubt will be very useful for policy development and future law reform efforts. I cannot speak to the rigour of the methodology/R analysis as this falls outside my area of expertise. Where I would like to see some additional consideration by the authors is in their concluding section. In particular, it was unclear to me why opt out rather than opt in is suggested for the central system for patients to control the uses of their data (“A reasonable balance might be a central system to opt out from identifiable clinical use, identifiable (s251) research use, or de-identified research use of one’s data, and to opt in for participatory research. This would complement efforts to improve people’s access to their own data.[74]”) The following paragraph states that ‘The majority support observed for clinical sharing without explicit consent perhaps makes such sharing reasonable as a default (opt-out) position, given the potential advantages for many people’s own care, subject to strict IG rules (who has access, when). However, a significant minority opposed this, mandating at least a public information campaign about opt-outs if this were to occur’. Given the opposition of ‘a significant minority’ (20% of respondents, and—for some demographic groups—considerably more than this), I would expect the authors to engage in a more nuanced way with the respective merits of an opt-in vs an opt-out system and potential policy implementation (e.g., the potential need for a ‘public good’ test to ground the sharing of information for research as proposed in Ballantyne, A., & Schaefer, G. O. (2018). Consent and the ethical duty to participate in health data research. Journal of medical ethics, 44(6), 392-396.)]
---

	I also would have liked to see greater links made with current legal requirements, including the General Data Protection Regulation (GDPR), the Data Protection Act 2018, and the duty of confidence that applies to data collected through clinical encounters. For example, under art 9.1 of the GDPR, only very limited conditions apply that allow data processing for 'personal data revealing racial or ethnic origin, political opinions, religious or philosophical beliefs, or trade union membership, and the processing of genetic data, biometric data for the purpose of uniquely identifying a natural person, data concerning health or data concerning a natural person's sex life or sexual orientation'. It is unclear how this would fit with the proposal for an opt-out sharing system. Given the public backlash that accompanied care.data, I also would have appreciated seeing more engagement with the lessons that can be drawn from that process.  • For example, how would patients be made aware of the opt-out system, especially noting that the most disadvantaged are the least likely to receive this information and the most likely to express concerns about sharing. Whose responsibility would this be and how could high enough uptake be assured? • Are there any forms of information that should be excluded from the centralised system (e.g., for care.data there were exclusions for HIV status, STIs, pregnancy termination, IVF treatment, marital status and some other sensitive information) Another issue that didn't appear to be mentioned in the manuscript is the possibility of the technology systems themselves not working (e.g., identifiers remaining in free text notes), which could severely impact trust.
--	---

REVIEWER	Colubri, Andres Harvard University , Organismic and Evolutinary Biology
REVIEW RETURNED	31-Oct-2021

GENERAL COMMENTS	The authors conducted the largest survey to date (29275 consenting participants) to assess public acceptance of sharing of health data with different destinations, including NHS for clinical care, academic, charitable, and private organizations for research. Without being an expert in the subject mater nor the survey and statistical techniques required to collect and analyze this data, I recognize the importance of such study in informing public policy that improves management of confidential MHI/PHI in accordance with public opinion. The manuscript includes an in-depth description of the survey instruments, descriptive statistic of the survey data, and several plots summarizing the main results, and I think that, thanks to the author's careful description of their methods, this study could be reproduced in the future with even larger number of participants. I have the general following comments: * Given the number of questions and groupings of participants, I wonder if and how multiple hypothesis testing approaches, such
--

	as FDR, should be applied to this context. One of the statements, "educational effects were relatively inconsistent", made me aware of this issue, what if some of the results are false positives due random variability? I feel we'd need a way of quantify that more precisely, but again, I may just be missing some basic element of survey analysis due to my lack of expertise in the area. * The authors warn that one of the weaknesses of the study is that, even the large sample size, "the sample remained under-representative of some demographic groups despite weighting." As far as I understand, this was an online survey that was disseminated by several health service sites, including 216 general practice surgeries and 154 large healthcare organizations. Given this apparently wide recruitment network, why certain demographics groups were still underrepresented, and what are ways in which future surveys could further increase participation of those groups. I think a more in-depth discussion of this aspect would be important to be included in the manuscript. * Effect of COVID-19 on the results. The authors say that support for data sharing increased during the pandemic. There were 8109 consenting participants prior to the UK lockdown in response to COVID-19, when such major and unexpected societal event takes place during the survey of this nature, I also wonder, from a methodological standpoint, what has to be done to control for the effect of this event. The impression I got from what was presented in the manuscript is that lockdown was not - how it was publicized
--	---

REVIEWER	Ivanova, Julia ASU
REVIEW RETURNED	06-Jan-2022

GENERAL COMMENTS	This study focuses on perceptions of UK residents regarding the sharing of personal health information (PHI) using three framing statements via a survey. Such large-scale studies regarding the topic are necessary for developing and implementing PHI-sharing technologies. The large sample size, though not representative of the UK population, is one of the study's biggest strengths. Comments: 1. Abstract and methods- Pandemic: Some mention of COVID-19 should be present within the abstract as the recruiting dates (Feb-Sep 2020) fall exactly at the very beginning of the pandemic in Europe through the first scientific reports regarding the virus. This is of import as public opinion regarding pandemic requirements and suggestions changed greatly toward the end of 2020 and then with the vaccine. Impact of the pandemic on this topic is significant as the sharing of PHI was brought into the forefront of public discussion as a way to help mitigate the viral spread. While the use of before and at/after lockdown split points were considered (methods, 4.8.), the changing dynamics of the public's understanding of the pandemic changed further after the end of recruitment. Conclusions are difficult to draw on a broader scale because of this. A follow up would help understand COVID-19
--

	impact on the research question and conclusions. As the factor pandemic was part of the model for all analyses, it is something to consider. 2. Abstract- A sentence about the type of methods used for expressed preferences would be ideal in design/setting/interventions/outcomes. Note, the use of “intervention” makes the reader think that you employ an active intervention (and as such, a specific method design), not gathering preferences: avoid the terminology or explain why this is considered an intervention. 3. Abstract-Results: Explain what “decreased with distance” might mean to the reader. 4. Introduction: p.3: Explain the statement that “‘care data’ have previously aroused public ire.” The statement “and for research also structured versus free-text (narrative) data” should be better explained. Note that free text is not necessarily narrative data, and in this case, should not be called that. In a few words define “framing” of data along with your citation. 5. Methods: 4.3. If a representative sample was a target, then there is methodology that is used to ensure its result. If specific steps were not taken toward this goal, I would recommend removing the discussion of a representative sample. 6. Methods: 4.9. The models appear logical and able to expound on the research questions. Considering the data, overall, I would also recommend the authors consider the impact of intersectionality when looking at demographics (certainly for any future projects or further analysis of the data). A far more thorough description of thematic analysis needs to be included here, though. Citations and steps for this method, along with what platform was used to code for thematic analysis and unit of coding should be included. At this time, this part of the study cannot be replicated without these steps. Therefore, results and outcomes from this method cannot be reviewed appropriately. 7. Results: 5.1. When discussing results such as “not everyone completed the survey” or majority/some/etc., please provide the number within the body of the paragraph and not just within the figures. For example, “primarily depression and anxiety disorders” should include the proportion or N since you then say “of them, 85%”... I would encourage the rewording of this section to ensure there is no confusion as to what you are referring to. The inclusion of numbers and percentages should be done throughout the results and discussion section (as well as within the abstract). 8. Results: 5.8. Results of a thematic analysis are usually better categorized and quantified. I would recommend considering Braun, V, Clarke, V. Using thematic analysis in psychology. Qual Res Psychol 2006 to help write out methods and display results effectively for the purposes of this article. 9. Figure 2: please include N for each of these items, especially knowing that not every participant provided this information.
--	---

VERSION 1 – AUTHOR RESPONSE

Reviewer #1: Dr David Wyatt, King's College London

Thank you for inviting me to review this well written and fascinating paper, which

We thank the Referee very much for their support and endorsement of our work.

draws on a vast amount of survey data to explore public opinion on data sharing for

health and research purposes. Focusing on data sharing without consent, the authors

document key new information in contemporary discussions of data sharing. Most

notably, differences between physical and mental health data, age, gender, ethnicity,

particularly in the context of sharing to the private sector.

While existing research has highlighted this general distrust and resistance to routine

health records being “sold” or utilised by the private sector, this study is the first I’ve

seen that adds texture to our understanding of the contours of these attitudes. Such

accounts are crucial if we are to explore ways of facilitating research while also

addressing the needs and concerns of the population.

I have provided comments below and some suggestions that I hope will help the

authors extend their analysis and discussion (on what is already a very good and

interesting paper).

Methods:

We apologize that we had not included this detail because; we are delighted to

expand on this. Because of other additions requested by the Referees, we are already

PPI involvement - It is clear from the author's list and the Contribution description

some way over the advisory word count so have added ~440 words describing the

that RAG members were heavily involved in this project. However the statement on

PPI processes in detail in the Supplementary Methods (S1.1), and referred to this in

pg 3 doesn't really do justice to this involvement - how was this arranged? How was

the Methods (4.1) along with some extra (but terse) detail here.

the study designed in practice? How were the different expertise of the group

managed to produce a socially robust but also academically sound survey? I'm

pushing here because it's listed as one of the strengths of the study and it's important

to demonstrate HOW PPI has helped and improved this process.

In terms of the quantitative analysis, I am not equipped to assess this meaningfully.

The design decisions, however, and questions asked seem reasonable, particularly the

inclusion different framings.

Results:	Yes; sorry about that. We have added some more detail to the Results text itself, as
These are clearly presented in text. I struggled a bit with the references to so many	requested by the other Referees and the Editor.
different tables and supplementary material items, but this is more a reflection on	
journal submission formatting criteria than the manuscript itself!	
9 Discussion:	Thank you; these have been very helpful. We have cited Carter et al. (2015) in the
Particularly in terms of the discussion (and possibly background), I think there is a lot	discussion of the concept of a social licence for health data processing (applicable
	likely beyond research to clinical practice also), and the others in relation to some of

of relevant work on data sharing for research in primary care (from a more social

science perspective) which highlights some of these issues and takes it further to

stress the need for trust, transparency and clear information governance. I think

engaging with some of the themes in these papers (I've included references to a

couple below), will help further develop the depth of the discussion.

Some of the themes in the literature cited below also seem to resonate well with the

free text answers (responding to the consent form) and discussed on Pg 8.

GREENHALGH, T., WOOD, G. W., BRATAN, T., STRAMER, K. & HINDER, S.

2008. Patients' attitudes to the summary care record and HealthSpace: qualitative

study. British Medical Journal, 336, 1290-1295.

HADDOW, G., BRUCE, A., SATHANANDAM, S. & WYATT, J. C. 2011. Nothing is

really safe': a focus group study on the processes of anonymising and sharing of

health data for research purposes. Journal of Evaluation in Clinical Practice, 17, 1140-

1146.

STEVENSON, F. 2015. The use of electronic patient records for medical research:

conflicts and contradictions. BMC Health Services Research, 15, 124.

You may also want to look at Carter et al. (2015) to see how we might conceptualise

the parameters of public acceptance of certain data sharing practices (discussed in the

context of care.data but relevant beyond this).

CARTER, P., LAURIE, G. T. & DIXON-WOODS, M. 2015. The social licence for

the subthemes raised by our participants in comments. We have expanded in

particular on the Conclusions (6.6) as set out below.

research: why care. data ran into trouble.
Journal of medical ethics, 41, 404-409.

10 In relation to Trusted Research Environments, Thank you. We have added the point about the ability
I can see how these would be an to exert control within TREs
important way of circumventing data control. (Discussion, section 6.6), and the principle of the “five
I think it’s important to also safes”, and cited the reference
acknowledge that the shift towards digital suggested (in its updated 21 July 2020 version, which
recording practices has meant that methods we think is the final one).
like TREs are key ways we can accurately control who has and does not have access
to certain data sets (and how they are used).
Outside of platforms like TREs, digital
data makes these kinds of controls so much harder. A key reference for TREs not
currently in the reference list is linked below:

UK HEALTH DATA RESEARCH ALLIANCE 2020. Trusted Research Environments
(TRE): A strategy to build public trust and meet changing health data science needs.

[https://ukhealthdata.org/wp-content/uploads/2020/04/200430-TRE-Green-Paper-
v1.pdf](https://ukhealthdata.org/wp-content/uploads/2020/04/200430-TRE-Green-Paper-v1.pdf)

I hope the authors find these comments helpful. Thank you once again for the We do indeed; thank you very much for these very helpful comments! opportunity to read this fascinating and important paper. I hope the authors find these comments helpful. Thank you once again for the We do indeed; thank you very much for these very helpful comments! opportunity to read this fascinating and important paper.

Reviewer #2: Dr Lisa Eckstein, University of Tasmania

12 This was a highly interesting dataset with nuanced information that no doubt will be very useful for policy development and future law reform efforts.

I cannot speak to the rigour of the methodology/R analysis as this falls outside my area of expertise.

We thank the Referee for their support and endorsement of our work, and their very helpful comments.

13 Where I would like to see some additional consideration by the authors is in their concluding section. In particular, it was unclear to me why opt out rather than opt in is suggested for the central system for patients to control the uses of their data (“A reasonable balance might be a central system to opt out from identifiable clinical use, identifiable (s251) research use, or de-identified research use of one’s data, and to opt in for participatory research. This would complement efforts to improve people’s access to their own data.[74]”)

The following paragraph states that ‘The majority support observed for clinical sharing without explicit consent perhaps makes such sharing reasonable as a default (opt-out) position, given the potential advantages for many people’s own care, subject to strict IG rules (who has access, when). However, a significant minority opposed this, mandating at least a public information campaign about opt-outs if this were to occur’.

Given the opposition of ‘a significant minority’ (20% of respondents, and—for some demographic groups—considerably more than this), I would expect the authors to engage in a more nuanced way with the respective merits of an opt-in vs an opt-out system and potential policy implementation (e.g., the potential need for a ‘public good’ test to ground the sharing of information for research as proposed in Ballantyne, A., & Schaefer, G. O. (2018). Consent and the ethical duty to participate in health data research. *Journal of medical ethics*, 44(6), 392-396.)]

14 I also would have liked to see greater links made with current legal requirements, including the General Data Protection Regulation (GDPR), the Data Protection Act 2018, and the duty of confidence that applies to data collected through clinical encounters. For example, under art 9.1 of the GDPR, only very limited conditions apply that allow data processing for ‘personal data revealing racial or ethnic origin, political opinions, religious or philosophical beliefs, or trade union membership, and the processing of genetic data, biometric data for the purpose of uniquely identifying a natural person, data concerning health or data concerning a natural person’s sex life or sexual orientation’. It is unclear how this would fit with the proposal for an opt-out	Thank you for this important point. The Referee is quite correct about these aspects of the EU GDPR (and its “legacy” form in UK law after the UK left the European Union) and the UK Data Protection Act. However, consent is emphatically not the legal basis by which the NHS holds confidential identifiable patient data. Patients often consent, of course—but sometimes they refuse, and have capacity to refuse, but their data is and must be held anyway (e.g. patients detained against their will under the Mental Health Act). There are extensive provisions in the GDPR and the Data Protection Act that permit data processing without consent, including of sensitive personal data, for a number of reasons. In many cases, opt-outs may be desirable but
---	---

sharing system.

are not legally required. Another key principle that applies is the “no surprises” rule

from Caldicott’s reviews as National Data Guardian. There is also legislation creating

a “duty to share” [Health and Social Care (Safety and Quality) Act 2015] between

health providers, where the subject does not object (and would not be “likely to

object”), so long as this does not conflict with data protection legislation.

We have set out our understanding of the highly complex topic of the legal bases, to

the best of our ability, in the Supplementary Materials (section S3.2; ~3,300 words)

and summarized it in the new Supplementary Figure 8. (Our apologies: we lost some

track-changes marks in the Supplementary Materials, but the whole of S3.2 is new.)

We think this covers all the points raised. We have added to the Discussion (much

more tersely!) about this.

15 Given the public backlash that accompanied care.data, I also would have appreciated We don’t have the answers to all of these questions but have noted the requirement to

seeing more engagement with the lessons that can be drawn from that process. engage particularly with such groups (Discussion, 6.6). We are not proposing any

- For example, how would patients be made aware of the opt-out system, new form of central data repository, merely a central system to express preferences,

especially noting that the most disadvantaged are the least likely to receive this but we take the point (also evident from the thematic analysis) that some respondents

information and the most likely to express concerns about sharing. Whose wanted to control the nature of information sharing in greater detail; we have added

responsibility would this be and how could high enough uptake be assured? to the Discussion (6.6) and suggest that this would require further consultation, if

- Are there any forms of information that such a central control system were to be implemented.

centralised system (e.g., for care.data there were exclusions for HIV status, STIs,

pregnancy termination, IVF treatment, marital status and some other sensitive information

16 Another issue that didn't appear to be mentioned in the manuscript is the possibility of the technology systems themselves not working (e.g., identifiers remaining in free text notes), which could severely impact trust.

Reviewer #3: Dr Andres Colubri, Harvard University, Broad Institute

17 The authors conducted the largest survey to date (29275 consenting participants) to assess public acceptance of sharing of health data with different destinations, including NHS for clinical care, academic, charitable, and private organizations for research.

Without being an expert in the subject matter nor the survey and statistical techniques

required to collect and analyze this data, I recognize the importance of such study in

informing public policy that improves management of confidential MHI/PHI in

accordance with public opinion.

The manuscript includes an in-depth description of the survey instruments, descriptive

statistic of the survey data, and several plots summarizing the main results, and I think

that, thanks to the author's careful description of their methods, this study could be

reproduced in the future with even larger number of participants.

We thank the Referee for their support, including of our methodological descriptions,

and their other helpful comments as below.

18 I have the general following comments:

This is an important and subtle point. We don't think that multiple comparisons are a

practical concern here. In this part of the survey, we collected data on (a) clinical, (b)

* Given the number of questions and groupings of participants, I wonder if and how multiple hypothesis testing approaches, such as FDR, should be applied to this context. One of the statements, "educational effects were relatively inconsistent", made me aware of this issue, what if some of the results are false positives due to random variability? I feel we'd need a way to quantify that more precisely, but again, I may just be missing some basic element of survey analysis due to my lack of expertise in the area.

research, and (c) linkage preferences. We analysed with one statistical model per dependent variable across all subjects, to give the overall picture, and then an additional single model per dependent variable across all subjects providing demographic information, to examine the effects of demographics. Analytically, we present significant terms from each model (Figure 6 with demographics, and Supplementary Figure 6 across the whole sample but without demographics). It is the norm to examine multiple terms in a single model without further multiple comparison correction. Where post-hoc tests are then conducted for significant factors with >3 levels, post-hoc corrections may be required. We mitigate against this firstly by not putting interpretational emphasis on specific pairwise comparisons for multi-level factors, and secondly by only presenting a subset of pairwise comparisons, those against a predefined reference category (e.g. as in Figure 6). We emphasize the uncorrected nature in the caption to Figure 6, and refer to the Supplementary Methods, where (in section S1.4) we discuss some of the statistical logic and the differences between the "omnibus" F test and the pairwise comparisons shown, and cite a further reference on this topic. This is imperfect—there is no one perfect way—but is helpful visually, allowing obvious patterns to emerge, such as the U-shaped preference curves observed for age. Additionally, many terms have such enormous effect sizes that their p value is infinitesimal (e.g. Figure 6). We discuss

this in the Supplementary Methods (S1.4), where we note also that we used simple

effects analysis for formal follow-up of significant interactions^a and contrast these to

the simple pairwise comparisons helpful for basic visual display.

We apologize that we may have given the wrong impression with the term

“inconsistent”. We have clarified what we meant by “inconsistent” in the Discussion

(6.4): that these specific effects (for education) were small and present for clinical

and linkage preferences, but in quantitatively different ways, whilst not being

significant for research preferences. Statistically, this is clear, but we would hesitate

to draw strong overall conclusions with respect to the effects of education given these

discrepancies.

^a Cardinal RN, Aitken MRF. ANOVA for the Behavioural Sciences Researcher. Mahwah, N.J: L. Erlbaum, 2006.

19 * The authors warn that one of the weaknesses of the study is that, even the large sample size, "the sample remained under-representative of some demographic groups despite weighting." As far as I understand, this was an online survey that was disseminated by several health service sites, including 216 general practice surgeries and 154 large healthcare organizations. Given this apparently wide recruitment network, why certain demographics groups were still underrepresented, and what are ways in which future surveys could further increase participation of those groups. I think a more in-	Yes; despite this large recruitment network of healthcare sites, there was significant bias. While the use of an online survey itself creates some biases, we think that a significant part of the reason is that the COVID-19 pandemic shut down the “face-to-face” component of network recruitment, which included recruiting staff being able to support participants directly if requested. The pandemic “lockdown” (and even after that, requirements to shield vulnerable groups) put paid to these plans. Essentially all NHS-supported research was put on hold, except studies
---	--

depth discussion of this aspect would be important to be included in the manuscript.

that were critical for the COVID-19 response itself (e.g. vaccination trials such as of the Oxford/AstraZeneca ChAdOx1 nCoV-19 vaccine, and treatment trials such as RECOVERY covering dexamethasone and other drugs, in which the NHS participated extensively) or could be conducted entirely online (such as our study). We attempted some measures (e.g. targeted advertising, creating an online video) but they were insufficient compensation. We therefore used weighting (raking) in sensitivity analyses, which confirmed our primary findings. We have added detailed discussion of these topics, and some potential future strategies. Mindful that we have added text elsewhere in response to the Referee's other points, and that we already exceed the recommended word limit significantly, we have placed this in the Supplementary Discussion (S3.1).

point 19.) Although much of the UK was following the news of terrible events in countries hit by COVID-19 earlier, the first event of stark national change was at the moment of first UK "lockdown". We give more detail on the exact sequence around that time in Chen 2020 a, but lockdown was an abrupt event, dramatically affecting the lives of nearly every UK citizen. We handled this analytically by splitting data into data from before

"lockdown" and that "at/after" lockdown (Methods, 4.8). As Referee #4 also points

out, public preference may have changed in more subtle ways during our study, and

beyond it. However, we think that testing more complex temporal models of the

pandemic's effects on preference, beyond the step change we tested for (and found

effects of), might well have low power to adjudicate beyond competing models, and

would take us far from our pre-planned protocol. We think that a single-variable

model is a reasonable minimal variation to the planned analysis that (empirically)

captured significant variance in response to a major unexpected event.

	a Chen S, Jones PB, Underwood BR, Moore A, Bullmore ET, Banerjee S, Osimo EF, et al. 'The Early Impact of COVID-19 on Mental Health and Community Physical Health Services and Their Patients' Mortality in Cambridgeshire and Peterborough, UK'. Journal of Psychiatric Research https://doi.org/10.1016/j.jpsychires.2020.09.020.
20 * Effect of COVID-19 on the results. The authors say that support for data sharing increased during the pandemic. There were 8109 consenting participants prior to the UK lockdown in response to COVID-19, when such major and unexpected societal event takes place during the survey of this nature, I also wonder, from a methodological standpoint, what has to be done to control fo the effect of this event. The impression I got from what was presented in the manuscript is that lockdown was not We agree: this was an unexpected and major event. We're not sure if some of the last sentence is missing, but the Referee is quite correct that the survey was not in any way publicized as being specifically in relation to the pandemic. (Recruitment methods, however, changed in response—as per the preceding - how it was publicized	

Reviewer #4: Ms Julia Ivanova, ASU

21 This study focuses on perceptions of UK residents regarding the sharing of personal health information (PHI) using three framing statements via a survey. Such large-scale studies regarding the topic are necessary for developing and implementing PHI-sharing technologies. The large sample size, though not representative of the UK population, is one of the study's biggest strengths.	Thank you for your kind comments on the work, and the very useful comments and suggestions below.
22 Comments: 1. Abstract and methods- Pandemic: Some mention of COVID-19 should be present within the abstract as the recruiting dates (Feb-Sep 2020) fall exactly at the very beginning of the pandemic in Europe	We agree entirely that this is a significant issue, and that preference may have changed further after the end of our study. We have added to the Discussion on this point, and on the limitations of our analysis (section 6.5). Mention of the COVID- 19-related results was already in the abstract, which has a very tight word limit, so we regret that we have not been able to expand

through the first scientific reports regarding the virus. This is of import as public opinion regarding pandemic requirements and suggestions changed greatly toward the end of 2020 and then with the vaccine. Impact of the pandemic on this topic is significant as the sharing of PHI was brought into the forefront of public discussion as a way to help mitigate the viral spread. While the use of before and at/after lockdown split points were considered (methods, 4.8.), the changing dynamics of the public's understanding of the pandemic changed further after the end of recruitment. Conclusions are difficult to draw on a broader scale because of this. A follow up would help understand COVID-19 impact on the research question and conclusions. As the factor pandemic was part of the model for all analyses, it is something to consider.	much on that point there. See also point 20 above.
23 2. Abstract- A sentence about the type of methods used for expressed preferences would be ideal in design/setting/interventions/outcomes. Note, the use of "intervention" makes the reader think that you employ an active intervention (and as such, a specific method design), not gathering preferences: avoid the terminology or explain why this is considered an intervention.	We agree that additional detail might be desirable but are unfortunately quite constrained by the 300-word limit (which we were already at). We have added detail regarding the Referee's point 24 (below) but space is very tight and further methodological additions would require cutting of results. We used an active experimental intervention, being the presentation of the framing statement—participants were randomized to receive one of three types of intervention—so we have retained that part of the heading.
Abstract-Results: Explain what "decreased with distance" might mean to the geographical distance reader.	Thank you. We have clarified that the meaning was literal, i.e. (as above), in the Abstract and the Results (5.4). We have also clarified (in 5.4) the more nuanced geographical effect seen when each destination and data type (nature) was enquired about and rated separately (Figure 3C) versus the opening one-from- many question about health data in general (Figure 2B).
25 4. Introduction: p.3: Explain the statement that "'care data' have previously aroused public ire."	We apologize that this was not clear. We have changed "public ire" to "public opposition and ire"; the cited reference (Sterckx 2015) describes in detail the response to this previous NHS England data pooling scheme, named "care.data" (care-dot-data) by them.

26 The statement “and for research also structured versus free-text (narrative) data” should be better explained. Note that free text is not necessarily narrative data, and in this case, should not be called that.	Thank you. We have expanded on this in the Introduction and cited further definitions of the two concepts, giving narrative information as one example of free-text data.
27 In a few words define “framing” of data along with your citation.	The framing is of decisions, not of the data; we apologize that this was not sufficiently clear. We have defined framing in more detail in the Introduction as requested, along with the citation to the Nobel prize-winning work on decision-making of Tversky & Kahneman (via their seminal 1981 review), of which framing was a central part.
5. Methods: 4.3. If a representative sample was a target, then there is methodology that is used to ensure its result. If specific steps were not taken toward this goal, I would recommend removing the discussion of a representative sample.	Our apologies. We have deleted “representative” as a target in the Methods (4.3), and retained discussion of the degree to which the actual sample was representative of the population (and in what ways it was not).
6. Methods: 4.9. The models appear logical and able to expound on the research questions. Considering the data, overall, I would also recommend the authors consider the impact of intersectionality when looking at demographics (certainly for any future projects or further analysis of the data).	Thank you for this very interesting point. Statistically, specific effects of intersectionality (the intersection of multiple characteristics or attributes, such as demographic characteristics, in an individual persona) are measurable and testable by way of interaction terms—that is, whether the effects of possessing characteristics X and Y together is different to what would be expected if their effects were simply additive (the sum of main effects).c,d We regret that despite the large data set, the than by specific sub-hypotheses) was too great. Specifically, our model C2 (analysis of “clinical” preference including the main effects of demographics and one interaction to test a specific hypothesis) had 27 terms (including one for the instruction to R to calculate a global intercept plus subject-specific random-effects intercepts). In contrast, a model with the same factors but allowing all interactions has 16,384 terms (and is not calculable on machines to which we have access since it requires >6.1 Tb

	RAM). Even allowing just all demographic interactions, without interactions between them and the primary analytical factors (destination, nature, detail, framing, pandemic), gives 1039 terms (and >127 Gb). For the research model, R2, we had 43 terms (calculated as above) but the full interaction model has 32,768 terms (and would require >54.7 Tb RAM), and the “full demographic interaction only” model has 1055 terms (>380 Gb). Computation times would likely also be infeasible. Even if these models were technically possible to solve, the number of hypotheses tested would be vast, and we think uninterpretable. Of course, we make our data publicly available so that those with specific interaction/intersection questions may test those. Our specific hypotheses related to main effects of demographics and the focused question of whether mental health experience affected willingness to share MH (versus PH) data. We have added to the Methods accordingly to expand on this point. a Crenshaw K. ‘Demarginalizing the Intersection of Race and Sex: A Black Feminist Critique of Antidiscrimination Doctrine, Feminist Theory and Antiracist Politics’. University of Chicago Legal Forum 1989, no. 1 (1989): 139–67. b Bauer GR, Churchill SM, Mahendran M, Walwyn C, Lizotte D, Villa-Rueda AA. ‘Intersectionality in Quantitative Research: A Systematic Review of Its Emergence and Applications of Theory and Methods’. SSM - Population Health 14 (2021): 100798. https://doi.org/10.1016/j.ssmph.2021.100798. c Cardinal RN, Aitken MRF. ANOVA for the Behavioural Sciences Researcher. Mahwah, N.J: L. Erlbaum, 2006.
--	---

	d Cardinal RN. 'Analysis of Variance'. In Corsini Encyclopedia of Psychology, edited by Weiner IB, and Craighead WE, 92–98. Hoboken, New Jersey: John Wiley & Sons, 2010	
30	A far more thorough description of thematic analysis needs to be included here, though. Citations and steps for this method, along with what platform was used to code for thematic analysis and unit of coding should be included. At this time, this part of the study cannot be replicated without these steps. Therefore, results and outcomes from this method cannot be reviewed appropriately.	We apologize that this information was lacking. We have added it to the Supplementary Methods (S1.5).
31	7. Results: 5.1. When discussing results such as “not everyone completed the survey” or majority/some/etc., please provide the number within the body of the paragraph and not just within the figures. For example, “primarily depression and anxiety disorders” should include the proportion or N since you then say “of them, 85%”... I would encourage the rewording of this section to ensure there is no confusion as to what you are referring to. The inclusion of numbers and percentages should be done throughout the results and discussion section (as well as within the abstract).	We have added numbers to the text as well as the figures in the Results, Discussion, and Abstract as requested, and clarified the section about experience of mental health conditions.
32	8. Results: 5.8. Results of a thematic analysis are usually better categorized and quantified. I would recommend considering Braun, V, Clarke, V. Using thematic analysis in psychology. Qual Res Psychol 2006 to help write out methods and display results effectively for the purposes of this article.	We apologize again for our failure to present this clearly. Detailed categorizations, descriptions, and tallies were already in the Supplementary Results (S2.9). We have not added the full thematic analysis to the main manuscript for reasons of space, but we have added tallies of common subthemes to the main results and made more explicit reference to the more detailed information. We have set out the process (see also point 30 above) with respect to the Braun & Clarke 2006 phases. We have also added more detailed cross-referencing to the Supplementary Material throughout the main manuscript.
33	9. Figure 2: please include N for each of these items, especially knowing that not	Thank you; yes, that's quite correct. Figure 2 already included absolute numbers of participants (bottom scale, “Number of

every participant provided this information.	participants”) as well as proportions (top scale, “Proportion of participants (%)”). We have added the totals to each panel to provide further clarity, mirroring subseque
--	--

VERSION 2 – REVIEW

REVIEWER	Eckstein, Lisa University of Tasmania , Faculty of Law
REVIEW RETURNED	02-Mar-2022

GENERAL COMMENTS	Thank you for this nuanced incorporation of reviewer comments. I read with interest and look forward to the published manuscript.
---